

# Why can people with developmental prosopagnosia recognise some familiar faces? Insights from subjective experience

Emma Portch[1], Liam Wignall[1,2] and Sarah Bate[1]

[1] Department of Psychology, Bournemouth University, Bournemouth, United Kingdom
[2] Department of Psychology, University of Brighton, Brighton, United Kingdom

## ABSTRACT

Developmental prosopagnosia is a relatively common visuo-cognitive condition, characterised by impaired facial identity recognition. Impairment severity appears to reside on a continuum, however, it is unknown whether instances of milder deficits reflect the successful use of spontaneous (typical) face recognition strategies, or the application of extraneous compensatory cues to recognition. Here, we explore this issue in two studies. First, 23 adults with developmental prosopagnosia were asked about their use of spontaneous versus compensatory face recognition techniques in everyday life, using a series of closed- and open-ended questions. Second, the same participants performed a computerised famous face recognition task where they were asked to provide reasons why they could make any successful identifications. Findings from both studies suggest that people with developmental prosopagnosia can successfully, and quite frequently, use compensatory strategies to recognition, and that these cues support the majority of instances of preserved familiar face recognition. In contrast, 16 of the 23 participants were able to spontaneously recognise familiar faces on at least some occasions, but there were vast individual differences in frequencies of success. These findings have important implications for our conceptualisation of the condition, as well as for diagnostic practice.

Corresponding author
Emma Portch,
eportch@bournemouth.ac.uk

## INTRODUCTION

Cases of prosopagnosia have long been reported in the academic literature. Traditionally these have been acquired cases where individuals with typical face recognition skills lose their face recognition ability following neurological trauma (*e.g.*, *Della Sala & Young, 2003*; *Valentine et al., 2006*; *Barton, 2008a*; *Bate et al., 2015*; *Corrow, Dalrymple & Barton, 2016*). In the majority of cases, acquired prosopagnosia is documented as a severe and permanent condition, where the key characteristic is an abrupt loss of the ability to recognise familiar faces (*e.g.*, *De Renzi et al., 1994*; *Barton, 2008b*). In the last 30 years, a more prevalent, developmental equivalent of this condition has emerged (*Bowles et al., 2009*; *Bennetts et al., 2017*; sometimes referred to as "congenital" or "hereditary" prosopagnosia, *e.g.*, *Behrmann*

& Avidan, 2005), presenting as an apparently lifelong inability to recognise faces in the absence of any concurrent developmental, neurological, intellectual, or psychiatric condition (Bate & Tree, 2017). As with acquired cases, the key symptom of developmental prosopagnosia (DP) that prompts people to contact researchers is an inability to recognise familiar others (Murray & Bate, 2018; Murray et al., 2018). However, this skill appears to be affected to differing degrees, and DP is commonly conceptualised as residing on a continuum of severity (Barton & Corrow, 2016; Bate & Tree, 2017).

While this notion of a spectrum is reasonable, it is complicated by the realisation that many DPs are able to successfully identify familiar faces using atypical mechanisms. These are often developed individually, outside of structured remediation training, with mixed and transient success that is dependent on recency of exposure and context (e.g., DeGutis et al., 2007; Bate & Bennetts, 2014). For instance, in a study that qualitatively probed such instances, researchers identified both interdependent compensatory strategies (e.g., identity prompts supplied both pre- and mid-event by others) as well as individual strategies, which centred around use of extra-facial cues (e.g., voice, hairstyle and gait, Adams et al., 2020). Such strategies may be particularly effective in people with DP compared to those who acquire prosopagnosia, because most will presumably have spent their lifetime developing and refining these elaborate strategies to assist with the recognition of others.

While identification and sharing of these strategies has provided a useful tool to help people with prosopagnosia cope with the condition (Adams et al., 2020), it remains unclear whether the same techniques may also account for apparently preserved familiar face recognition that has been observed experimentally in people with DP. That is, the variability in recognition that drives the conceptualisation of DP as a spectrum may simply reflect the differential application and success of compensatory skills, rather than evidencing differences in face recognition ability per se. Alternatively, it may be that some DPs are able to "naturalistically" or "spontaneously" recognise a small number of familiar faces, at least with no conscious use of compensatory strategies (hereafter referred to as "spontaneous" recognition). This may occur because the person has a limited (but intact) face recognition capacity, perhaps aided by additional circumstances (e.g., over or very recent exposure to a particular face, the distinctiveness of the face, or an emotional connection to the owner). In these instances, recognition may be slower or subject to more rapid decay than in typical perceivers but may nevertheless be supported by typical processing mechanisms. Clarifying whether apparently preserved familiar face recognition is underpinned by naturalistic mechanisms versus compensatory strategies would therefore have important implications for our conceptualisation of DP and variations in its severity.

In addition, understanding the reasons behind apparently preserved familiar face recognition is important in screening and diagnosis. Most laboratories confirm DP using a range of unfamiliar face recognition tests, but these are often supplemented by a "famous" face recognition test (Barton & Corrow, 2016; Dalrymple & Palermo, 2016; Bate & Tree, 2017). Such tasks typically involve the identification of a set of celebrity faces, which may be cropped to reduce the use of extraneous cues to recognition (e.g., Duchaine & Nakayama, 2005; Duchaine, Germine & Nakayama, 2007; Bennetts et al., 2015; Arizpe et al., 2019; Bate et al., 2019a). However, when familiar faces are recognised in these tests, many

DPs ascribe their recognition experience to the specific picture used, rather than the face itself, particularly when stimuli are frequently seen and thus regarded as iconic (*Duchaine & Nakayama, 2004*; *Carbon, 2008*; *Bennetts et al., 2015*; *Murray & Bate, 2020*). Similarly, DPs sometimes suggest that recognition was driven by attention to a specific and / or distinctive feature, rather than the whole face (*e.g.*, a beauty spot; *Dalrymple & Palermo, 2016*). Given these instances of recognition contribute to the total correct score on famous face tasks, it is unsurprising that the vast majority of DPs recognise at least some celebrity faces. In fact, in one such dataset, atypical famous face scores of those with confirmed DP ranged from 10.00 - 73.68% correct, with two further DPs achieving scores within the typical range (*Bate et al., 2019b*). This wide variation in DP performance on famous face recognition tasks has been replicated in other studies using different stimuli and participant samples (*e.g.*, *Stollhoff et al., 2011*; *Mishra et al., 2021*). It is therefore important to elucidate whether instances of successful identification in these tests reflect a genuine ability to spontaneously recognise particular familiar faces, or the application of elaborate compensatory strategies. Such findings may have important implications for current diagnostic guidelines regarding the interpretation of DP performance at screening, and consequently our overall understanding of the presentation of the disorder.

Here, we address this issue in two studies. Our overall aim was to develop understanding of familiar face recognition by people with DP, including the reasons for successful and unsuccessful performance. While many studies have attempted to examine this issue from an experimental visuo-cognitive perspective (*e.g.*, by examining underpinning processing strategies *via* experimental paradigms or methodologies such as eye-tracking, *e.g.*, *Bennetts et al., 2022*), here we took an experiential viewpoint to gather subjective information about performance from the DPs themselves. First, we examine every day familiar face recognition in DP. Because this cannot easily be replicated in an experimental setting, we used a questionnaire containing closed- and open-ended questions to probe experiences of using compensatory and spontaneous face recognition strategies in everyday life. Second, we addressed the same issues in a computerised famous face recognition task, where DP participants were asked to identify images of celebrities and, where appropriate, to indicate the reasons *why* they were able to recognise that individual. Thus we examined the generalisability of findings from Study 1, and addressed the question of whether these factors also assist performance in an objective task that is typically used for prosopagnosia screening. While it is not common practise either to impose or generate specific, testable hypotheses from work with an experiential focus (Study 1), the frequency-based findings from Study 1 led us to predict that compensatory recognition would significantly inflate DP performance on the famous face task in Study 2.

## STUDY 1

An initial study aimed to investigate the use of compensatory *versus* spontaneous recognition strategies by people with DP in the everyday recognition of familiar faces. To probe this issue, we developed an online questionnaire composed of closed- and open-ended response options.

## METHOD

### Participants

A total of 23 participants (20 female, 3 male; $M_{age} = 46.7$ years, $SD_{age} = 10.6$, range = 22–59) with an existing diagnosis of DP took part in this study. All had previously self-referred to our group reporting everyday difficulties with face recognition, despite normal or corrected-to-normal vision, and the absence of any neurodevelopmental disorder, neurological damage or psychiatric illness. Following existing protocols (*e.g.*, *Dalrymple & Palermo, 2016*; *Bate & Tree, 2017*) all participants were screened for DP using objective tests of face recognition ability (see Table S1). The study adhered to the ethical standards outlined in the Declaration of Helsinki and set forward by the British Psychological Society and ethics approval was granted by the Institutional Ethics Committee at Bournemouth University, UK (approval reference: 32530). All participants provided their informed written consent to participate.

### Materials and Procedure

Participants completed an online questionnaire (*via* the Qualtrics survey platform; see full questionnaire presented in Article S1; data are available in *Data S1*) that comprised two sections regarding their (1) use of compensatory cues to recognition, and (2) ability to spontaneously or automatically recognise faces. There were six questions in each section. Nine questions required the selection of one frequency-related response on a five-point scale. For instance, the first section enquired about the frequency with which participants use compensatory strategies for recognition and the success of these strategies; whereas the second section enquired about the consistency and success of participants' ability to spontaneously recognise familiar faces. These closed-ended questions were supplemented with three open-ended questions that invited participants to reflect on the types of error that they make with each type of strategy.

### Statistical analyses

For all closed-ended questions, the number of DPs that selected each response option was summed. The three open-ended questions were analysed by the second author, using thematic analysis (*Braun & Clarke, 2006*), with the first and third authors verifying identified themes against the data set.

## RESULTS

### Section 1: Compensatory cues to recognition

An initial set of questions enquired about the use of compensatory cues to recognition (*e.g.*, when spontaneous face recognition fails but the participant can "work out" a familiar person's identity using extraneous cues such as context, hairstyle, gait or accessories; see Fig. 1 for specific answer frequencies, per question). Participants were asked whether they are able to successfully use these cues to recognise familiar faces: although no participant indicated that they are able to successfully use these cues to recognise all faces that they should know, all of the DPs stated that they can be used effectively for the recognition of at least a few individuals, with equal modal frequencies indicative of recognising *many*
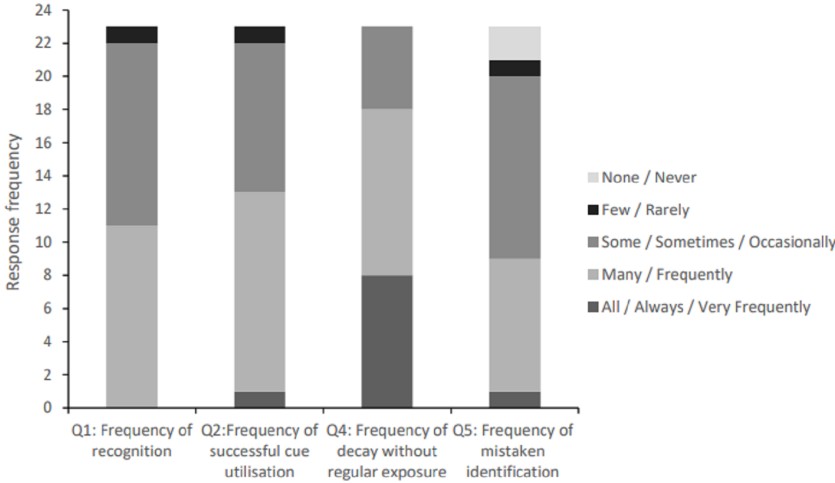

**Figure 1** **Question response frequencies for everyday instances of compensated recognition (N = 23).** *Note.* Response options varied by question, with the following format adopted in Question 2 and 4: 'Always', 'Frequently', 'Sometimes', 'Rarely' and 'Never'. Instead, Question 1 posed the following options: 'All', 'Many', 'Some', 'Few' and 'None'; and Question 5: 'Very Frequently', 'Frequently', 'Occasionally', 'Rarely' and 'Never'.

or *some* faces this way, and only one participant selecting the *few* option. Pertinently, no participant indicated a complete inability to use compensatory strategies for the recognition of any familiar individual.

Participants were then asked if these strategies always result in successful recognition of the relevant familiar faces. Only one participant indicated that they can *always* recognise these faces without error, with the majority of participants indicating that such recognition attempts are *frequently* or *sometimes* successful. One participant *rarely* recognises faces successfully *via* the use of compensatory strategies, but no participant indicated that this method of recognition entirely fails.

The 22 participants who responded that compensatory strategies fail in at least some instances were then presented with an open-ended question inviting them to free-type their reflections on why these failures occur. Most participants emphasised the importance of context in helping them recognise faces; when context (*e.g.*, work environment) was taken away, participants struggled to recognise people. For example, P2 said, "If I see a person out of context, I am much less likely to be able to recognise them." Similarly, P6 said, "I use context a lot. If someone is out of context, I can't find the right cues to look for to identify them." P19 said their strategies fail "When people are not in the context, almost always." Finally, P20 said, "Often [strategies fail] because of a context change, like seeing a co-worker outside of work or when I'm not expecting to see them."

Participants also described using distinguishing features to recognise people—primarily their hairstyles. Consequently, when people changed their hairstyles, participants found it more difficult to recognise them. P7 said they struggle to recognise people due to "Sometimes obvious things, like people change hairstyles or wear a different style of clothes." Similarly, P8 said, "The most common reason is that people change their

hairstyle or colour." Highlighting how both context and distinguishing features were used in combination, P1 said, "My strategies fail when clues like context are taken away, or when people change their hairstyle, or if I can't hear them speaking", while P2 said, "If I see a person out of context, I am much less likely to able to recognise them. Also, people change their hair, facial hair and make-up!"

Finally, some participants stated that some people do not have sufficiently characteristic or distinctive features for participants to recognise them. For example, P7 said, "Sometimes people's distinguishing features aren't distinct enough for a 100% match every time." Similarly, P14 said, "Anyone with similar features is likely to identified as the same person."

This question was supplemented by a further closed-ended question that enquired about the longevity of successful use of compensatory strategies for identity recognition. Modal responses indicated that these abilities *always* or *often* decay when participants do not regularly see the person, with fewer participants indicating that the ability *sometimes* decays. No participant suggested that the ability *rarely* or *never* decays.

The final question in this section asked whether the use of compensatory strategies can result in false alarms (*i.e.,* the mistaken thought that an unknown person is 'familiar'). While one participant stated that this happens *very frequently*, modal responses indicated that this was a *frequent* occurrence, or happened only *sometimes*. Fewer participants stated that false alarms happen *rarely* or *never*. The 22 participants who experience false alarms on at least some occasions were invited to free-type their reflections on why these errors occur. The main rationale given was that strangers shared the physical features of people participants knew. For example, P1 said:

> I may see a person who fits a particular mental picture of someone I know, based on how I remember their hairstyle, body shape and clothing... When I think of people, I know I usually picture their hair as the first thing I think of.

Similarly, P2 said, "People wearing clothes and having hair that's similar to someone I do know can confuse me." Finally, P6 said, "Sometimes the person has the right hairstyle and age, so you assume it is a particular person, but it isn't." Participants' strategies predominantly involved using non-facial features to help recognise a person, which subsequently resulted in false alarms when encountering other individuals who share those features, with P18 stating they make mistakes when "The stranger falls into the rough bracket I've made for the person I know."

Participants described the process of recognising people as trial and error, with P6 saying, "Identifying someone is a case of informed guessing. Sometimes you guess wrong." Similarly, P4 said, "When you are looking for someone you are not sure you will recognise, it is easy to slot the wrong person in." There was some apprehension about incorrectly identifying a stranger as somebody they knew, with P2 saying, "To be honest, I rarely make the first move anymore as I'm too anxious about making mistakes."

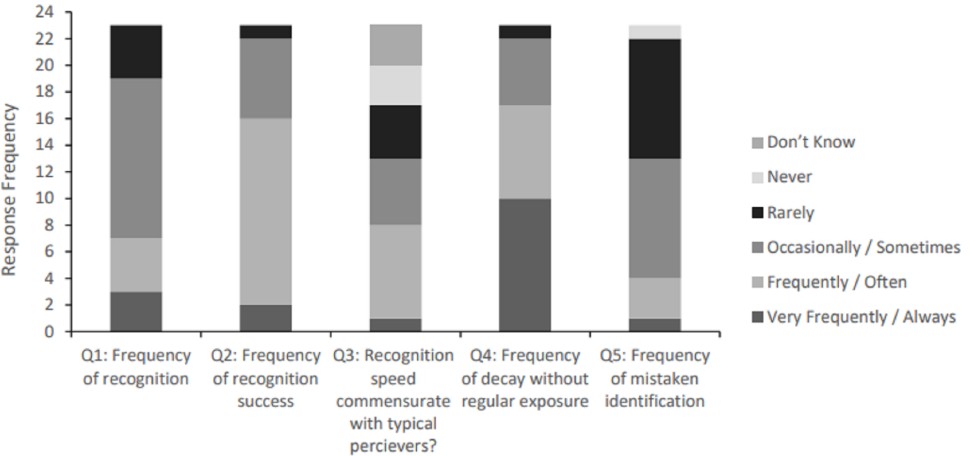

**Figure 2** **Question response frequencies for everyday instances of spontaneous recognition (N = 23).** *Note.* Response options varied by question, with the following format adopted in Question 1 and 5: 'Very Frequently', 'Frequently', 'Occasionally', 'Rarely' and 'Never'. Instead, Question 2–4 posed the following options: 'Always', 'Often', 'Sometimes', 'Rarely' and 'Never', with a "Don't Know" option added for Question 3, only.

## Section 2: Spontaneous recognition of familiar faces

The next set of questions enquired about the "spontaneous" recognition of familiar faces, utilising core facial information alone and without the need to apply compensatory or contextual cues (see Fig. 2 for specific response frequencies, per question). The modal response suggested that participants *occasionally* recognise faces spontaneously, while fewer responses indicate that the ability is *very frequent*, *frequent* or *rare*. No participant responded that they can *never* perform this skill.

Participants were then asked about their frequency of success for spontaneous recognition of faces that were at least sometimes recognised that way. While very few participants indicated that this type of recognition was successful on *every* occasion or was *rare*, most participants indicated that they could *often* perform this task successfully, with intermediate response frequencies suggesting that this process was *sometimes* successful.

The next questions enquired about the limitations of spontaneous recognition. In terms of latency, only one participant felt that they could *always* spontaneously recognise faces with the same speed as people without face recognition difficulties. The modal response frequency suggests that participants could *often* recognise faces as quickly, with fairly equal and intermediate response frequencies given to the other question options (*sometimes, rarely,* and *never*). Three participants stated that they *don't know* if they recognise faces as quickly as typical perceivers.

The next question enquired whether the ability to spontaneously recognise a face can decay without frequent exposure to that person. While the majority of participants indicated that this ability *always* or *often* decays if they do not regularly see a person, fewer participants suggested that decay *sometimes* occurs, and one *rarely* experiences decay. No participant reported the complete absence of decay without recurrent exposure.

Finally, participants were invited to reflect on false alarms that might occur during spontaneous face recognition. One person indicated that they *very frequently* mistake an unknown person for a familiar one, while modal response frequencies suggested that false alarms happened *sometimes* or *rarely*. Fewer participants suggested that these errors happened *frequently* or *never*. The 22 participants who make false alarms during spontaneous recognition on at least some occasions were asked to free-type the reasons why they think these failures occur. Participant responses partially mirrored those given to the open-ended question about failures in compensated recognition scenarios.

A minority of participants could not articulate why these errors occurred, with P19 stating they were "unsure" and P18 saying, "the recognition is so quick that I'm not sure why it went wrong." P18 added, "I would guess that my mind has taken the environment, face and other factors into account and matched a person that I usually recognise well, and I haven't done any manual processing to check that the identification is correct."

A common rationale given by participants was that the person had similar characteristics to the person they thought they recognised. For example, P11 said the person, "Looks similar to the person I thought it was. Same face shape / age / sex / colour and similar hairstyle." P14 similarly highlighted that confusion arose due to, "Similar features shared between two people", with P12 also specifically emphasising, "Similar hairstyles, clothing."

Participants also mentioned that people within the same family often shared facial features, and this sometimes underpinned incorrect spontaneous recognition experiences. For example, P8 said errors happen with people, "When they are members of the same family, or when they just happen to have similarly-shaped faces" with P7 also saying, "It would mostly be family and very close friends - occasionally I might see someone who looks like one of them, and I know that usually, given the context, it's unlikely to be." P3 complicated this further:

> I am an idiot when it comes to recognition. More nuancedly, sometimes two people just look identical to me…And people who are supposed to look identical, that is others say they look identical, do not always look identical to me. For example, one of my family members had an identical twin, but I had absolutely no difficulty telling those two apart –I didn't even think they looked similar, though everyone else said they did.

Finally, despite typically being considered a compensatory cue to recognition, participants identified context as contributing towards some errors in spontaneous recognition. For example, P6 said, "I rely on context. I might expect a person to be in a place or on a photo, so I spontaneously recognise the wrong person. My expectation is a stronger influence than recognition skills." Similarly, P10 said, "misleading context cues" play a role, while P7 said, "It happens less frequently the other way around, *i.e.,* seeing someone I'd expect to recognise automatically and not recognising them. That's only happened a handful of times, and always when I've seen the person hugely out of context." Finally, P4 identified context as the cause of the errors, alongside expectations of potentially seeing the person, saying, "You are looking for someone you expect to be there and someone vaguely similar can easily be imagined to be the right person."

P20 was the only participant to mention the impact of face masks (worn to prevent the transmission of COVID-19), saying:

> It doesn't happen often but something that has been happening recently is seeing people with / without masks is causing me to make more errors. If people look similar (for example, they're related) and I get that recognition feeling with them, I have recognised their face I've just failed to remember the person I have recognised.

### Summary of Study 1

All DP participants indicated that they are capable of using compensatory strategies to aid recognition, and they use these strategies frequently and often successfully. However, these strategies are subject to decay without repeated exposure to familiar individuals. They cannot be used when a person is met out of context, changes their appearance (*e.g.*, hairstyle), or simply does not possess distinctive features. False alarms can also occur when an unknown person shares physical features with a familiar person (*e.g.*, the same hairstyle).

All participants indicated that they can spontaneously recognise familiar faces, but that compensated routes are more often used. When spontaneous recognition does occur, many participants report that recognition is often successful, although open-ended responses indicated that it is often assisted by context, itself regarded as a compensatory cue. Responses varied on the latency of spontaneous recognition, but most agreed that the ability decays without frequent exposure to a face. False alarms were reported to occur infrequently and were driven by instances where an unknown person looks physically similar to a familiar person.

In sum, conclusions drawn from the everyday experiences of people with DP indicated that familiar faces can sometimes be spontaneously recognised, and also offer a variety of explanations for why compensatory recognition can sometimes occur. These were taken forward into a second study that examined their use in an objective face recognition test that is frequently used for prosopagnosia screening.

### Study 2

Study 1 indicated that successful familiar face recognition in everyday life by people with DP is more frequently supported by the use of compensatory rather than spontaneous processing strategies, although the two were not directly compared and were difficult to disentangle (*e.g.*, the use of context, considered a compensatory cue, was still an important implicitly or explicitly-applied aid for spontaneous recognition). A second study aimed to more objectively examine the same issue, by asking DP participants *how* they successfully recognised familiar famous faces during a recognition task. We reasoned that if performance on successful trials followed the experiential pattern identified in Study 1 (*i.e.,* that correct identifications are more frequently supported by compensatory than spontaneous recognition), typically-derived accuracy scores, which conflate the two, may obscure more profound difficulties in face recognition ability. This would have important implications for DP screening.

## METHOD

### Participants

The same DP participants that were used in Study 1 were resampled for Study 2. A total of 49 control participants (31 female, 18 male) were also recruited for the study, aged between 30 and 65 years ($M_{age} = 45.2$, $SD_{age} = 10.8$). All had lived within the UK for their entire life. No participant reported any history of neurological, psychiatric, visual or developmental conditions. The study adhered to the ethical standards outlined in the Declaration of Helsinki and set forward by the British Psychological Society and ethics approval was granted by the Institutional Ethics Committee at Bournemouth University, UK (approval reference: 32530). All participants provided their informed written consent to participate.

## MATERIALS

The famous face recognition task comprised 200 images: 100 famous identities and 100 non-famous distractors. Our inclusion of distractors is a departure from most DP screening paradigms where only celebrities are presented for identification (*e.g.*, *Duchaine & Nakayama, 2005*; *Duchaine, Germine & Nakayama, 2007*; *Arizpe et al., 2019*; *Mishra et al., 2021*). By balancing the experimental design according to familiarity we eliminated the potential for participants to assume that every face should be identifiable, thus reducing extraneous inflation to successful recognition rates and increasing the need for discriminability.

The faces of 100 famous identities were selected (35 female—familiarity was favoured above gender balance; see the full list of celebrity names provided in Article S1) from a government survey that probed individuals whom the UK public considered to be the most 'famous' or well-known (https://yougov.co.uk/ratings/entertainment/fame/people/all). One image of each celebrity was subsequently selected from a name-based search using Google images; the image that was the first (top) result in the search, per identity. If the face image in that search position was of poor quality, side-facing, or partially occluded, it was deemed unsuitable, and replaced with the image in the next position. High-quality images of distractor identities, who could plausibly be perceived to work in the entertainment industry but were not famous, were selected from an actor's 'extras' webpage. Distractor identities were matched to each celebrity according to gender, age, and perceived attractiveness, with these determinations made by the first and third authors. This resulted in a pool of 200 images: 100 celebrities and 100 matched distractors. The images were split into four blocks of 50, each containing 25 celebrities and 25 distractors.

All images were cropped below the chin, without excluding any external features (*i.e.,* the full head, including the hair, was visible). While other famous face tasks have presented tightly cropped, internal-feature only stimuli, we wanted to include the full head to mirror ecologically valid instances of everyday face recognition where this visual information is nearly always available (*e.g.*, *Burton, 2013*; *Murray et al., 2021*). In keeping with this rationale, image background was not fully standardised, though we selected images where

the background was not able to provide cues to identity. All images were adjusted to 400 pixels in height, but the width was permitted to vary to prevent image distortion.

## Procedure

Participants completed the task remotely, *via* the Qualtrics platform. The order of the four blocks was randomised for each participant, and all trials were randomised within each block.

In each trial, the target face was displayed at the top of the screen and remained there until the participant indicated whether the face was familiar or not. Participants progressed immediately to the next trial if the face was unfamiliar. If the face was thought to be familiar, the image disappeared and the participant was asked to either provide a name for the identity, or a differentiating piece of information that demonstrated their knowledge of the target (*e.g.*, a film they had appeared in, a song they had performed, a programme they had presented, or a public role they had filled).

DP participants were then asked to provide reasons for why they were able to recognise the face. They viewed a number of response options: 'I just have a vague feeling of familiarity when I look at this face'; 'I have seen this particular image before'; 'this face has a distinctive feature'; 'this overall face is particularly memorable'; 'I have seen this face enough times to memorise it'; 'I have strong emotional feelings (positive or negative) towards this person, and this helps my recognition' and 'this is a face that I am able to spontaneously recognise (*i.e.,* in an "instant" or "automatic" manner where I know who the person is without having to "work it out")'. For each face, participants could select as many responses as necessary and were also given the opportunity to free-type any additional reasons for recognition. These steps were also presented when distractor stimuli were incorrectly judged to be familiar.

Familiar face recognition purportedly occurs rapidly and "automatically" for typical perceivers (*e.g.*, *Jung, Ruthruff & Gaspelin, 2013*; *Zimmermann, Yan & Rossion, 2019*) and thus may be associated with a relative inability to consciously unpack and articulate the steps involved (cf. controlled cognitive processing; *e.g.*, *Schneider & Shiffrin, 1977*). As such we did not collect this same data from control participants as they would be unlikely to provide meaningful and accurate responses to what they might consider a confusing question. This asymmetrical design also complemented our study aims, which were to explicitly examine the underpinnings of atypical recognition in a group of people with a qualitatively different experience of face recognition in daily life. Thus, the role of control participants in this study was simply to provide normative baseline data for the performance (accuracy) measures that were observed in the famous face recognition task (see Table 1).

Finally, all participants viewed the names of all celebrity faces used in the task and were asked to rate their familiarity with the person on a scale anchored from 1 (not at all familiar) to 5 (highly familiar). This allowed us to remove any celebrities from the task that participants were generally unfamiliar with, irrespective of their face recognition difficulties.

**Table 1** Average percentage accurate (SD) for Controls and DPs in each task component.

|  | Control $M$ ($SD$) | DP $M$ ($SD$) |
| --- | --- | --- |
| Target Familiarity (Hits) | 95.25 (4.29) | 50.65 (22.07) |
| Distractors (Correct Rejections) | 96.80 (3.77) | 92.83 (8.62) |
| Correct Target Identifications | 89.38 (9.63) | 41.65 (19.72) |

## Statistical analyses

For completeness, overall performance on the task is reported for DP compared to control participants. The proportion of celebrities correctly recognised was adjusted for each participant, accounting for any that the participant judged as 'completely' or 'mostly' unfamiliar when presented with their name.

To address the study's specific research questions, responses were initially separated across all DP participants into famous *versus* distractor trials. Because we were interested in reasons for recognition, trials were separated for each DP participant according to those famous faces that were (a) correctly categorised as familiar but were not able to be identified, (b) correctly identified either by name or unique semantic information, and (c) correctly categorised as familiar but were subsequently misidentified by name or by the provision of semantic information that was instead indicative of a different (incorrect) celebrity. Distractor trials were separated for each participant into those that were (a) incorrectly categorised as familiar but without a name / semantic information offered for the identity, and (b) misidentified as a specific celebrity (*via* provision of either an incorrect name, semantic information, or both). For each of these response categories, we calculated (a) the base frequency, and (b) the proportion with which each of the reasons for recognition (see above) were provided. Frequencies and proportions were summed and averaged respectively, across all participants. Few additional reasons were entered. Where this did occur, all but four were reallocated to the "distinctive feature" option, as they merely identified the exact feature involved. Famous trials that were missed (incorrectly designated to be distractor faces) and distractors that were correctly rejected were not analysed.

Inferential analyses were conducted on averaged proportional endorsement rates, only, which were calculated per reason for recognition across instances of successful categorisations, identifications, spontaneous and non-spontaneous (compensatory) recognitions (*i.e.,* 26 proportional totals, per DP participant; see Data S2). Bonferroni-adjusted *p*-values were used to infer statistical significance while lessening the potential for Type 1 errors; differential alpha criteria are reported, per analysis, below. All reported *p*-values are two-tailed.

## RESULTS

Overall scores for basic recognition performance were initially calculated for DPs and controls (see Table 1 and Data S2). Based on this data, 21 DPs (91.30%) achieved scores that were more than two standard deviations below the control mean on the key identification

**Table 2   Summed frequencies of reason-for-recognition endorsement rates given by DPs across all task components.**

|  | Sufficient exposure | Spontaneous recognition | Vague sense of familiarity | Overall face is memorable | Distinctive feature | Image Recognition | Emotion-driven recogniton |
|---|---|---|---|---|---|---|---|
| *Famous:* | | | | | | | |
| Categorisation | 0 | 0 | 94 | 1 | 8 | 11 | 0 |
| Identification | 656 | 300 | 194 | 368 | 330 | 160 | 75 |
| Misidentification | 9 | 0 | 16 | 8 | 3 | 2 | 1 |
| *Distractor:* | | | | | | | |
| False categorisation | 0 | 0 | 84 | 1 | 10 | 47 | 3 |
| Misidentification | 7 | 0 | 31 | 1 | 3 | 4 | 0 |

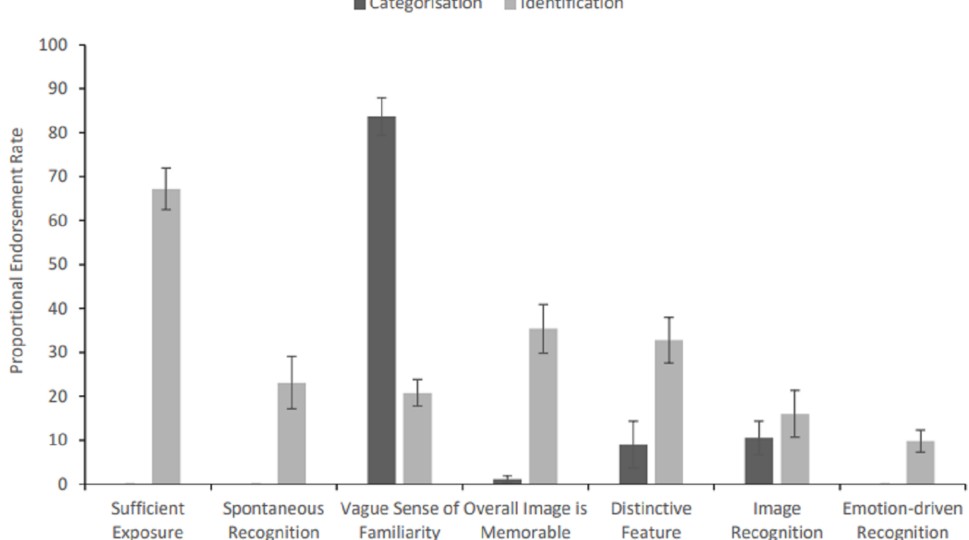

**Figure 3   Proportional reason endorsement rates for successful instances of famous face categorisation and identification, respectively (error bars represent SEM).** *Note.* Participants were able to choose multiple reasons for each instance of recognition, thus proportional endorsement rates may exceed 100% overall for both categorisation and identification.

measure that is typically used in screening (*e.g.*, *Duchaine, Germine & Nakayama, 2007*; *Bennetts et al., 2015*).

For famous faces that were successfully recognised (either by categorisation as 'familiar' only, or with provision of differentiating identifying information *e.g.*, a name or semantic information, see above), the summed frequency of occasions that each reason was selected, by response category, is summarised in Table 2, and the averaged proportion of reason endorsement, as a function of the total number of responses in each category, are provided in Fig. 3. Two additional reasons for recognition were also cited: for three famous trials where the target was correctly identified, participants stated that they had seen that face in the few hours preceding their participation. One participant also stated that they correctly identified a famous face because the person had an unusual name that they were able to associate with the face.

For famous faces that were correctly categorised as familiar but were not identified, the reason for the response was predominantly that the participant had a vague sense of familiarity when they saw the face. Other reasons that attracted lower endorsement frequencies were having seen the image before, judging that the overall face was memorable or that it had a distinctive feature. For famous faces that were correctly identified (by either name or unique semantic information), the primary reason by far was that the participant had sufficient past exposure to the face which facilitated recognition of newly encountered images. This was supplemented by all the other reasons (see Table 2 and Fig. 3), but particularly aided by the overall face or a distinctive feature being memorable.

A 2 (type of recognition: categorisation only *vs.* successful identification) $\times$ 7 (reason for recognition: see above) within-subjects ANOVA was conducted on proportional averaged endorsement rates, which returned main effects of type of recognition ($F$ (1, 21) = 40.27, MSE = 409.83, $p < .001$, $\eta_p^2 = .66$), reason for recognition ($F$ (6, 126) = 27.61, MSE = 408.16, $p < .001$, $\eta_p^2 = .57$), and a two-way interaction ($F$ (3.77, 79.24) = 64.82, MSE = 421.57, $p < .001$, $\eta_p^2 = .76$, Greenhouse-Geisser correction applied). To explore the interaction, first a one-way ANOVA was conducted on proportional reason endorsement rates for cases of categorisation, only: $F$ (6, 126) = 102.29, MSE = 202.17, $p < .001$, $\eta_p^2 = .830$. Bonferroni-corrected *post-hoc* tests (base alpha criterion of .05 / 21 comparisons = .002; N = 22) revealed that a vague sense of familiarity was endorsed a significantly higher number of times than any other reason for this type of recognition (all $ps < .001$). A second one-way ANOVA also revealed significant differences in proportional reason endorsement rates for instances of successful identification, $F$ (6, 132) = 17.84, MSE = 464.24, $p < .001$, $\eta_p^2 = .45$. Bonferroni-corrected *post-hoc* tests showed that sufficient exposure to the face was implicated in successful identifications significantly more often than vague feelings of familiarity, spontaneous, emotion- or image-driven mechanisms (all $ps < .001$). Reasons of overall memorability or the face having a distinctive feature held an intermediate position, with their endorsement rates neither differing significantly from those associated with significant exposure ($ps \geq .003$), nor vague feelings of familiarity, spontaneous, emotion- or image-driven mechanisms ($ps$: .010–1.00).

Paired samples *t*-tests were also used to compare proportional averaged reason endorsement rates across successful cases of categorisation and identification; a Bonferroni-corrected alpha criterion of .007 was applied to these comparisons (*i.e.*, $p = .05 / 7$ comparisons, N = 22). While a vague sense of familiarity was endorsed a significantly higher number of times for cases of categorisation (cf. identification; $t$ (21) = 12.78, $p < .001$, $d = 2.72$), all other reasons were endorsed at a significantly higher rate for cases of identification (cf. categorisation; all $ps \leq .002$, $ds \geq .77$), with the exception of image-driven mechanisms, which were endorsed comparably across both types of successful recognition, $t$ (21) = 1.16, $p = .257$, $d = .25$.

We were also particularly interested in successful recognition attempts that DPs indicated were spontaneous in nature. Supporting findings from Study 1, these recognition instances accounted for a substantial, but smaller, number of identifications than those achieved *via* compensatory mechanisms ($M = 23.40\%$ per participant of that individual's correct identifications, $SD = 28.77$, see Fig. 4). There was also much higher variability in the number

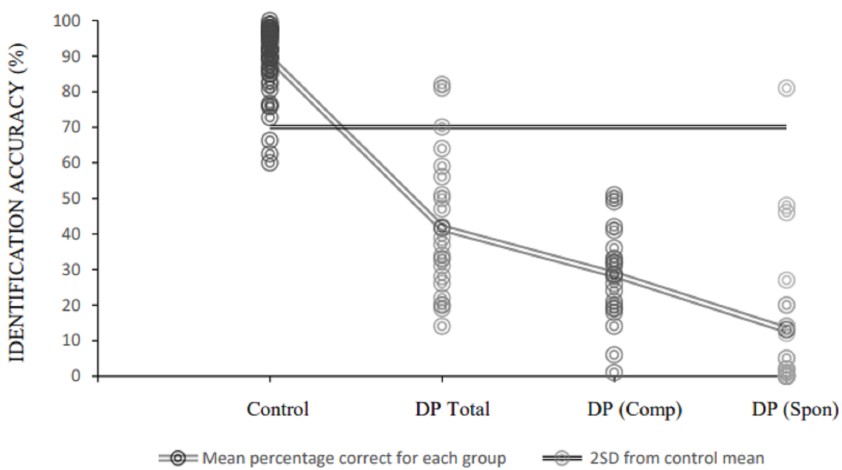

**Figure 4  Accurate famous face identification percentages for control and DP participants.** Individual DP accuracy rates are plotted according to (A) total correct identifications, (B) correct identifications attributed to compensatory mechanisms, only, and (C) correct identifications achieved spontaneously. Trend lines indicate (A) average identification accuracy, per group (dark grey), and (B) two standard deviations below the control mean of 71.20% (black).

of successful identifications attributed to spontaneous *versus* compensated recognition, with use of the former recognition route ranging from 0.00 –98.78% in the sample. While two DPs attained recognition accuracy within two standard deviations of the control mean when identifications attained *via* both routes were considered, only one participant reached this benchmark based on spontaneous identifications, alone. Indeed, when examining the impact on proportion correct by removing trials that did not represent spontaneous instances of recognition, the average decrease in score was 28.61% ($SD = 13.12$, range = 1–51%), resulting in a mean performance of 13.04% ($SD = 20.45$).

Only four participants indicated that they achieved spontaneous recognition for some identities without implicating additional reasons for recognition (3.33% of all cases of successful spontaneous recognition). The three most common accompanying reasons were sufficient exposure to the face (see Fig. 5), perceiving that the overall face was memorable or had a particularly distinctive feature. Fewer participants indicated that they had seen the particular image before or had strong emotional feelings about the depicted identity. A 2 (type of recognition: spontaneous *vs.* non-spontaneous) × 6 (reason for recognition: see above) within-subjects ANOVA conducted on averaged proportional reason endorsement rates returned a significant main effect of reason for recognition ($F (5, 75) = 13.50$, MSE = 944.15, $p < .001$, $\eta_p^2 = .47$), a non-significant main effect of type of recognition ($F (1, 15) = 1.78$, MSE = 927.28, $p = .202$, $\eta_p^2 = .11$), and a significant two-way interaction ($F (5, 75) = 8.58$, MSE = 399.19, $p < .001$, $\eta_p^2 = .36$). To explore the interaction, first a one-way ANOVA was conducted on proportional reason endorsement rates for cases of spontaneous recognition, only: $F (5, 75) = 16.60$, MSE = 729.98, $p < .001$, $\eta_p^2 = .53$. Bonferroni-corrected *post-hoc* tests (base alpha criterion of $p = .05$ / 15 comparisons = .003) revealed that sufficient exposure to the face was the modal response and accrued
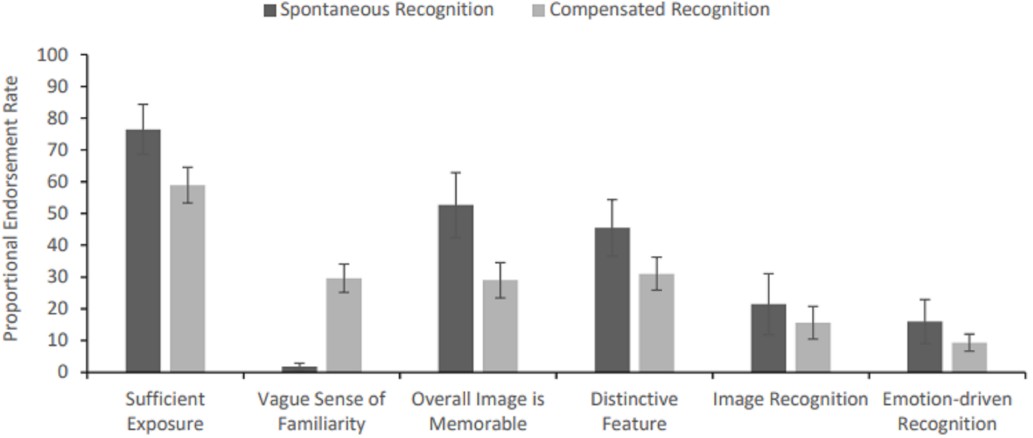

**Figure 5  Proportional reason endorsement rates for spontaneous and compensated successful recognitions (categorisation and identification rates are collapsed per bar; error bars represent SEM).** *Note*: Participants could endorse multiple reasons per successful recognition attempt, thus total summed proportional endorsement rates for spontaneous and compensated instances of recognition may exceed 100%, respectively.

significantly higher endorsement rates than a vague sense of familiarity, or image- and emotion-driven mechanisms (all $ps \leq .001$), but not overall memorability ($p = .664$) nor distinctiveness ($p = .030$), themselves holding intermediate, and similar, endorsement rates ($p = .950$). A vague sense of familiarity was the least frequently endorsed reason, attracting significantly fewer selections than overall memorability and distinctiveness ($ps = .003$), but comparable selections with image- and emotion-driven mechanisms ($ps \geq .925$).

A second one-way within-subjects ANOVA was conducted on averaged proportional reason endorsement rates for compensated instances of recognition, $F(2.89, 63.78) = 11.75$, $MSE = 988.83$, $p < .001$, $\eta_p^2 = .35$ (Greenhouse-Geisser correction applied). Participants again endorsed sufficient exposure to the face as the modal reason, which attracted more frequent selections than the two least-endorsed reasons; image- or emotion-driven recognition ($ps \leq .002$), which themselves attracted similar endorsement rates ($p = .958$). Reasons of overall memorability, distinctiveness and vague familiarity were endorsed at an intermediate, and statistically indistinguishable rate ($ps \geq .925$).

Paired samples $t$-tests were also used to compare proportional endorsement rates across instances of spontaneous and compensated recognition, respectively, with a Bonferroni-corrected alpha criterion of .008 applied (*i.e.*, $p = .05 / 6$ comparisons; N = 16). Here only a vague sense of familiarity attracted a significantly higher endorsement rate for compensated *versus* spontaneous cases of recognition, $t(15) = 6.36$, $p < .001$, $d = 1.59$. All other reasons were endorsed at a statistically indistinguishable rate across the two types of recognition ($ps \geq .047$, $ds \leq .55$).

Misidentifications for famous faces were primarily driven by vague feelings of familiarity (see Fig. 6) that may have been erroneously linked to a different individual. Non-negligible proportional endorsement rates were also observed for reasons of sufficient exposure to the 'face' and perceiving the overall face as memorable. Vague feelings of familiarity remained
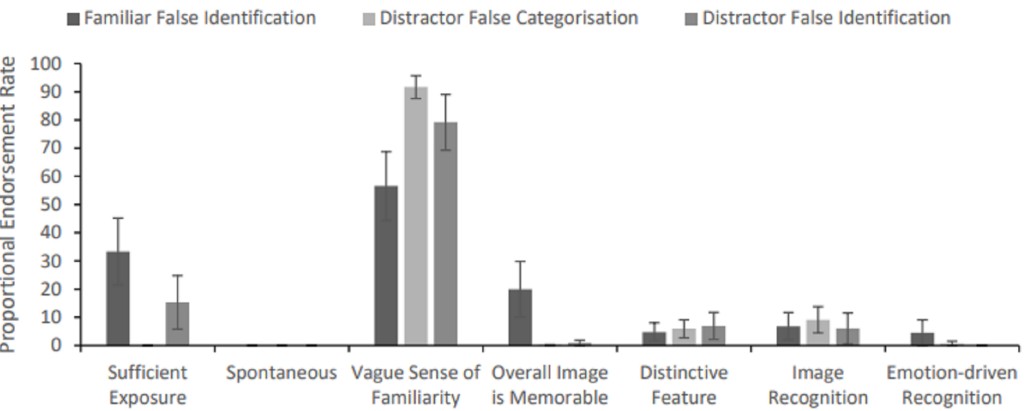

**Figure 6** Proportional reason endorsement rates accompanying (A) false identification of familiar famous faces, (B) false categorisation of distractor faces, and (C) false identification of distractor faces (error bars represent SEM). *Note.* Participants could endorse multiple reasons per erroneous recognition attempt, thus total summed proportional endorsement rates per type of erroneous response may exceed 100%.

the key cited reason for falsely indicating that a distractor was familiar, with participants also suggesting that they recognised the specific image or that the face had a particularly distinctive feature. For distractor images that were incorrectly identified (either *via* name, or semantic information) a vague sense of familiarity was again the modally endorsed reason for recognition, with a smaller number of false alarms also associated with sufficient exposure, image recognition or judged distinctiveness of the face. It was not possible to inferentially compare proportional endorsement rates across instances of false alarms as so few participants had made all three types of error (N = 6).

## Summary of Study 2

Findings from this study indicate that spontaneous recognition mechanisms account for approximately 23% of correct identifications of famous faces in DP participants, although there are vast individual differences in this figure (see Fig. 4). Indeed, seven of the 23 participants were not able to spontaneously recognise any famous faces, whereas four others used spontaneous recognition for only <10% of their successful identifications. This suggests that compensatory recognition techniques are driving many of the instances of preserved familiar face recognition in the condition, as well as artificially heightening scores on famous face recognition tasks that are used for diagnosis. Nevertheless, spontaneous recognition is possible for most people with DP, despite the range in frequency of success. Addressing experiences that support recognition, data suggest that vague feelings of familiarity are frequently linked to either erroneous recognition attempts (for familiar and unfamiliar faces), or partial successful recognitions (*i.e.,* where participants were simply able to categorise the face as 'familiar' but could not name or semantically describe the individual). In contrast, successful instances of (predominantly spontaneous) full identification are arguably driven by stronger experiences; in particular, sufficient exposure to the face, with overall or feature-based memorability also receiving intermediate

endorsement rates. Interestingly these reasons were more often implicated in identification than recognition of a specific facial image, which has been suggested to inflate recognition rates in previous famous face tasks that have used similarly iconic images (*e.g.*, *Bennetts et al., 2015*; *Murray & Bate, 2020*).

## GENERAL DISCUSSION

This investigation aimed to examine the reasons why people with DP are able to successfully recognise some familiar faces. Across two studies, DPs were asked about their use of compensatory and spontaneous face recognition strategies in (a) everyday settings, and (b) a computerised famous face recognition task. Findings from both studies indicate that instances of successful familiar face recognition in DP are primarily supported by the application of compensatory techniques which are often successful. Nevertheless, most (but not all) DPs are capable of spontaneous recognition on some occasions, although there are vast individual differences in the extent of this ability. These findings have important implications for our conceptualisation of DP, and for diagnostic practice.

First, the findings that compensatory cues to familiar face recognition are used frequently, and often successfully, fit well with previous work that advocates the identification and sharing of these strategies as tools to assist people with the condition (*e.g.*, *Murray et al., 2018*; *Adams et al., 2020*; *Murray & Bate, 2020*). The qualitative findings reported here further extend existing knowledge by demonstrating that these strategies can readily be disrupted in everyday life when people appear outside of an expected context, when they change distinguishing aspects of their appearance (*e.g.*, hairstyle or hair colour), or when the person simply does not possess sufficiently 'distinctive' features to support recognition.

Second, findings across both studies suggest that most DPs are capable of spontaneously recognising familiar faces on at least some occasions. In Study 1, where questions enquired about this ability in everyday life, it emerged that such instances were often qualified by context (whether implicitly or explicitly considered), and spontaneous recognition is much less successful when familiar people are unexpectedly encountered. While context is typically seen as a recognition aid, it is of course intertwined with spontaneous recognition even for typical perceivers. Study 2 enabled us to examine the ability more objectively, where familiar celebrities were intermixed with unknown distractors for recognition. Here, we calculated that only 23% of all successful identifications of celebrities across DP participants could be attributed to spontaneous recognition, and there was considerable variation in capacity between individuals.

It was common for participants to implicate additional reasons when they judged recognition to be spontaneous (96.67% of cases) and arguably the relative endorsement frequencies of each may provide vital clues to what drives these preserved abilities. While erroneous recognition attempts and simple correct categorisations were primarily accompanied by a vague sense of familiarity, successful spontaneous recognition was associated with arguably stronger experiences; most notably that the face had been frequently seen but, to a lesser extent, that the overall face was memorable or possessed a distinctive feature. That DPs commonly require sufficient exposure to a face may suggest

that spontaneous recognition abilities are built in a cumulative way and may follow a swift decay function if repeated exposure is not possible. However, sufficient exposure was also the modally implicated reason for successful compensated recognitions, with no significant difference observed when comparing proportional endorsement frequencies for each recognition type. Thus, sufficient exposure may also be important for learning which compensatory strategies might be most applicable for a particular face to differentiate it from others. Indeed, this would support the results of Study 1 as participants suggested that both spontaneous and compensatory forms of recognition followed a swift decay function without repeated and recent exposure to a face. High endorsement frequencies for overall memorability and distinctiveness of the face may also suggest that spontaneous recognition is stimulus-dependent and not possible for all faces.

These findings have important implications for our conceptualisation of DP. For the individual experiencing the condition, the most striking characteristic is arguably the failure to recognise familiar others—particularly the faces of close family and friends (*Murray et al., 2018*; *Murray & Bate, 2020*). While this is often portrayed as an absolute aspect of the condition in acquired cases (*Valentine et al., 2006*; *Bennetts et al., 2017*), the DP literature often refers to a continuum of severity in a conceptualisation that is not dissimilar to other developmental disorders (*Barton & Corrow, 2016*; *Bate & Tree, 2017*). What has remained unknown to date is whether instances of preserved familiar face recognition might be supported by spontaneous recognition experiences (akin to those demonstrated by typical perceivers) or the application of compensatory strategies. The work reported here suggests that compensatory cues assisted with successful identifications in as many as three-quarters of trials across participants, and in everyday life is often supported by context. Given that DP is suspected to be a lifelong condition where individuals will develop elaborate recognition strategies over time, it may be prudent to adjust our conceptualisation of the condition to acknowledge that the key symptom is a difficulty in recognising familiar faces when encountered out of context. Pertinently, when famous face recognition scores were adjusted to only include instances of successful spontaneous recognition, data continued to support the premise that DP resides on a continuum of severity.

Importantly, these findings have critical implications for diagnostic practice. Instances of successful recognition using compensatory techniques inflated famous face recognition scores by an average of approximately 29% across all the DP participants. Future work should investigate whether there is a need to partial out such successful identifications, or whether famous face tasks have ample sensitivity to correctly identify individuals with DP without such an adjustment. Practically, famous face screening tasks can readily be adapted to enquire exactly *how* successful identifications were made, in the same manner as reported here, with the proportion correct adjusted accordingly.

Our findings also have further implications for DP screening. Indeed, the accuracy of self-reported face recognition ability during screening has been much debated, with many authors pointing to only a mild relationship between objective and subjective diagnostic measures (*e.g.*, *Tree & Wilkie, 2010*; *Palermo et al., 2017*; *Arizpe et al., 2019*; *Murray & Bate, 2020*; *Estudillo & Wong, 2021*; but see *Shah et al., 2015*; *Gray, Bird & Cook, 2017*; *Tsantani, Vestner & Cook, 2021*). The findings reported here might offer a potential

means of reconciling this literature: people with face recognition difficulties may differ in their accuracy of self-report because they are unaware of the efficacy of their supporting strategies as opposed to their face recognition ability itself. Further, while we have probed compensatory recognition of familiar faces in this paper, it is also likely that some DPs are able to apply a different set of compensatory techniques to the learning of unfamiliar faces, and differences in the success of these techniques might assist only some individuals on objective screening tasks such as the Cambridge Face Memory Test (CFMT; *Duchaine & Nakayama, 2006*).

## CONCLUSIONS

The key finding reported here is that most DPs still exhibit some preserved capacity for familiar face recognition. Study 1 revealed that most DP participants associated both spontaneous and compensatory recognition with moderate to high success rates, and a swift decay function if exposure to the face had not been sufficient or recent; a finding mirrored in the proportional averaged reason endorsements observed in Study 2. Both studies also revealed that the presence of distinguishing features was important for both types of recognition, and qualitative responses in Study 1 showed that false alarms could arise when features were modified, perhaps reducing their effectiveness as a unique identifier. Findings also suggest that, as with typical perceivers, context is intrinsically important to recognition attempts and that, while both forms of recognition were modally associated with the same reasons for recognition, successful spontaneous attempts are infrequently associated with vague feelings of familiarity or false alarms, perhaps suggesting that spontaneous recognition is experienced by the individual differently to compensated recognition. Observations of vast individual differences in recognition ability, particularly for spontaneous instances, support the current conceptualisation that DP relies on a continuum of severity. We therefore suggest that core definitions of the condition should reflect these findings, and updates to diagnostic practice would further inform understanding. Critically, if the field can further elucidate the mechanisms that preserve some familiar face recognition in the condition, this could drive the development of remediation techniques that might further improve these processes.

### Funding

Sarah Bate is supported by a Leverhulme Research Fellowship (RF-2020-105). The funders had no role in study design, data collection and analysis, decision to publish, or preparation of the manuscript.

### Grant Disclosures

The following grant information was disclosed by the authors:
Leverhulme Research Fellowship: RF-2020-105.

## Competing Interests

The authors declare there are no competing interests.

## Author Contributions

- Emma Portch conceived and designed the experiments, performed the experiments, analyzed the data, prepared figures and/or tables, authored or reviewed drafts of the article, and approved the final draft.
- Liam Wignall analyzed the data, authored or reviewed drafts of the article, and approved the final draft.
- Sarah Bate conceived and designed the experiments, performed the experiments, analyzed the data, prepared figures and/or tables, authored or reviewed drafts of the article, and approved the final draft.

## Human Ethics

The following information was supplied relating to ethical approvals (i.e., approving body and any reference numbers):

Bournemouth University's Institutional ethics panel granted approval for the project under reference code: 32530. Project approval was granted in June, 2020.

## Data Availability

The data is available in the Supplemental Files.

## Supplemental Information

Supplemental information for this article can be found online at http://dx.doi.org/10.7717/peerj.15497#supplemental-information.

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
