# Peer review of "Why can people with developmental prosopagnosia recognise some familiar faces? Insights from subjective experience"

_PeerJ, doi:10.7717/peerj.15497_

## Round 0.1 · original submission · Major Revisions

Thank you for your submission to PeerJ. I have now received reviews from two experts, who thought there was some promise in your work, but both recommended 'major revisions' before the manuscript might be suitable for publication. Reviewer 2 had particular issues with the first study, while Reviewer 1 had some concerns about the second study. You can see all of the comments below, and these need addressing before your manuscript can be considered for publication. Thank you for submitting your work, and all the best with tackling these reviews. I look forward to reading the revised version of the manuscript upon its completion.

Reviewer 1 ·

Basic reporting

Clear and concise

Experimental design

Mostly sound, but see 4.

Validity of the findings

Mostly sound, but see 4.

Additional comments

This is a well written and clearly structured paper. I would like to commend the authors on their use of free-response options to investigate the psychological processes of DP's - there is great value to be found in simply asking people about their experiences, especially when the field is so focused on variants of face recognition/memorability tests that have less than perfect reliability and validity.

Study 1 was clearly written and reported, but Study 2 was unclear. I felt the authors could have more clearly stated the rationale for Study 2 - in particular, are the authors having participants give responses to the identification strategies that were identified in Study 1? It felt like this was the logical step but it was not made clear how Study 1 informed Study 2 and its methods.

It is also unclear what is going on with the central analysis of Study 2. The authors used Friedman's ANOVA and Wilcoxon Signed Rank tests here, but there's not much need to really go down that route; for the most part parametric statistics are quite robust to non-normality assumptions. You are also unlikely to see a truly normal distribution in the data for Study 2 as Figure 6 highlights the DV is a proportion/accuracy, which is naturally bounded. This data could be reanalysed with a clearer approach, i.e. ANOVA or t-tests. The authors also mention a control group in the introduction to Study 2, and they appear in Figure 6, but its not clear what role the controls serve. The tests in the analysis do not explicitly make reference to them so its not clear what purpose the analysis is serving. Did the controls also highlight their mechanisms for identifying faces? Should a comparison be drawn between the controls and the DP's here more explicitly? Are DP's using any novel mechanisms that controls don't use, or at least use them at different rates to controls?

In general, Study 2 could be made much clearer with a more principled analytical approach and clearer aims.

Reviewer 2 ·

Basic reporting

The manuscript explores the theme of how familiar faces are recognized in individuals with congenital prosopagnosia through direct questions on the use or not of compensatory cues in everyday life and during the performance of a task of familiarity and famous face identification.
The work is interesting and useful for thinking about the possibility and design of rehabilitation tools for these individuals, however, it has some weaknesses.

a. A main issue is the implicit association made by the authors between the consciousness about our ability to recognize faces and the performance obtained. My experience in recruiting and assessing individuals with congenital prosopagnosia suggests that some persons do not even know to have atypical performance in face recognition and others do think to have difficulties while they have average scores on face recognition tests. A big group of researchers, working in different labs and having the same experience, published a paper about the low correlation between self-assessment and objective assessment (Palermo et al., 2017). Moreover, strategic compensation in adults can be completely automatized and spontaneously used as it happens in patients with homonymous hemianopia that show spontaneous oculomotor compensation strategies (e.g., Zihl J. Oculomotor scanning performance in subjects with homonymous visual field disorders. Vis Imp Res. 1999;1(1):23–31; Elfeky A, D'Août K, Lawson R, Hepworth LR, Thomas NDA, Clynch A, Rowe FJ. Biomechanical adaptation to post-stroke visual field loss: a systematic review. Syst Rev. 2021 Mar 27;10(1):84). Furthermore, like visual field defects, the strategies implemented spontaneously may not even be optimal (Kerkoff et al, 1992; Zihl, 2011). Finally, the self-assessment of why a face has been recognized could be a post-hoc inference, and not necessarily a correct metacognitive knowledge. Given all these possibilities, direct questions about self-performance and the conscious use of compensatory cues can be very interesting and informative but cannot answer the question of the title: “Why can people with developmental prosopagnosia recognize some familiar faces?”


Consequently, I suggest the following changes:
1) Modify the title. Something including the adjective “ subjective” is more appropriate to the type of measures.
2) Modify the term “spontaneous recognition” with “no conscious use of compensatory cues” (vs “conscious use of compensatory cues) or, as an alternative, discuss the limitation of the construct.
3) Include the criteria and the tests scores for the diagnosis of each individual with congenital prosopagnosia to give an idea of their difficulties severity and characteristics, given the heterogeneity described in the literature and that also reported by the authors. Table 1 is not about this.
4) The selection criteria can have an impact on the results, particularly in the case of the self-reports, generating a bias. So, the scores to validated tests of face recognition are necessary to verify the correlation between the subjective indexes and the objective performances in validated tests.

b. In the paper the hypotheses are not well defined. The possible use of compensatory cues and different perceptual mechanisms is not new (e.g., Adams et al., 2020; Dalrymple & Palermo, 2016), and the alternative hypothesis proposed suggests a memory deficit: “In these instances, recognition may be slower or subject to more rapid decay than in typical perceivers but may nevertheless be supported by typical processing mechanisms.” (p.8 lines 89-90). To answer to the alternative hypothesis an objective test of face memory would be more appropriate. On the other hand, a lot of research have showed as DP do not show typical processing mechanisms. A proof of atypical mechanism is the inversion effect, not present in DP and thought to be an index of global configural processing of faces (e.g., Behrmann M. & Avidan G., 2005. Congenital prosopagnosia: face-blindfrom birth. Trends in Cognitive Science, 9,180–187; Cattaneo, Z., Daini, R., Malaspina, M., Manai, F., Lillo, M., Fermi, V., ... & Comincini, S. (2016). Congenital prosopagnosia is associated with a genetic variation in the oxytocin receptor (OXTR) gene: an exploratory study. Neuroscience, 339, 162-173.).

c. Another issue is about the term “developmental prosopagnosia”. Although it is used in many papers, many findings show that there is a genetic marker of this condition (e.g., Cattaneo et al., 2016; Susilo T, Duchaine B. Advances in developmental prosopagnosia research. Curr Opin Neurobiol. 2013;23(3):423–429; Kennerknecht I, Pluempe N, Welling B. Congenital prosopagnosia--a common hereditary cognitive dysfunction in humans. Front Biosci. 2008 Jan 1;13:3150-8; Schmalzl L, Palermo R, Coltheart M. Cognitive heterogeneity in genetically based prosopagnosia: a family study. J Neuropsychol. 2008 Mar;2(1):99-117) and, more, the participants of the two studies described by the authors are all adults. Then, the term “congenital prosopagnosia” (Behrmann & Avidan, 2005) is more appropriate then “developmental prosopagnosia”.

Experimental design

In the first study, the authors described the development of an online questionnaire composed of closed- and open-ended responses. How can they know whether the questionnaire is valid and reliable? How do the different questions are independent or not? Why did they not make use of the PI20 questionnaire (Shah et al., 2015) to verify at least the convergent validity?
No strong conclusions can be drowned by that study.
The second study presents an interesting task, whose strength lies in having also introduced the faces of unknown characters to evaluate the response bias. Nevertheless, instead of presenting data as the accuracy, hits, and false alarms, they should compute sensitivity (d') and response bias or criterion indexes derived by the signal detection theory (Tanner, Wilson P.; Swets, John A. (1954). "A decision-making theory of visual detection". Psychological Review. 61 (6): 401–409) to take full advantage of the presence of non-famous faces.

Validity of the findings

See previous comments

Additional comments

Minors
The sentence “Developmental prosopagnosia is a relatively common visuo-cognitive condition” (abstract p 6 line 26) is meaningless considering that the cut-off in neurologically healthy individuals is statistically determined by the mean and SD of the normal distribution.

To sum up, the authors should add the assessment of the face recognition difficulties, change the title and the hypothesis in terms of “subjective” judgments about the conscious use of compensatory strategies in CD, and soften the conclusions.

---

## Round 0.2 · Minor Revisions

As you can see, Reviewer 1 is now satisfied with your revisions but Reviewer 2 very much isn't. I was hoping that you could (better) address Reviewer 2's comments here, and then I can make a decision on whether to send it back out to Reviewer 2 or not.

Reviewer 1 ·

Basic reporting

Great

Experimental design

Great

Validity of the findings

Great

Reviewer 2 ·

Basic reporting

See additional comments

Experimental design

See additional comments

Validity of the findings

See additional comments

Additional comments

I am not satisfied with the revision: the authors have not made any major changes other than the title. Also, now that they added the scoreboard to facial recognition tests, 1.7 SD below the controls' average is a questionable criterion. Already Duchaine and Nakayama in the 2006 work on the CFMT used the criterion of 2 sd and wrote: "neuropsychologists often classify scores two standard deviations below the mean as compromised..." (2006 p. 582). Furthermore, three individuals (P14, P21, and P23) still do not show the DP criterion (two out of three "pathological" scores) and I don't understand why they included those subjects in the sample.

---

## Round 0.3 · accepted · Accept

I would like to thank you for your patience in this process. As you know Reviewer 1 was previously satisfied with your revisions, while Reviewer 2 was not. After your attempts to improve the manuscript further, I invited Reviewer 2 to again review your work. However, they have failed to provide a review on this round (after accepting the invitation) and so I have decided to accept your work since I am also satisfied that you have dealt with the comments well.